# I am where I believe my body is: The interplay between body spatial prediction and body ownership

**Francesca Frisco[1,2,3], Vito Bruno[1], Daniele Romano[1,2,3], Giorgia Tosi[1,2,3]***

**1** Department of Psychology, University of Milan-Bicocca, Milan, Italy, **2** Milan Center for Neuroscience, Milan, Italy, **3** Mind and Behavior Technological Center, University of Milan-Bicocca, Milan, Italy

* giorgia.tosi@campus.unimib.it

## Abstract

Body ownership refers to the feeling that the body belongs to oneself. This study explores how our ability to predict our body's location in space influences feelings of ownership and disownership towards it, comparing two illusion techniques: the virtual Rubber Hand Illusion (vRHI) and the first-person perspective Full-Body Illusion (1pp-FBI). Participants were exposed to each illusion, where they observed a virtual body aligned or misaligned with their own. Participants observed the virtual body for 60s (i.e., visual exposure) and then experienced the multisensory body illusion. During the illusion, participants received tactile stimulation while observing the avatar being synchronously touched. After two minutes, a virtual knife appeared and stabbed the virtual body. We recorded the Skin Conductance Response (SCR) as an implicit embodiment measure. After the visual exposure and the body illusion, we administered a Body Localization Task to evaluate the body's perceived position and a questionnaire to measure embodiment and disembodiment subjective experience. We performed a series of Bayesian regression in a factorial within-subject design. Results showed that both illusions increased feelings of ownership, but this effect was weaker in the misaligned 1pp-FBI. Interestingly, disownership only occurred in the misaligned 1pp-FBI, particularly when the legs were misaligned. Additionally, we found a recalibration of the body's position toward the misaligned virtual body, but no changes emerged when the real and the fake body were aligned. Correlation analyses partially supported the hypothesis that the perceived body's position influences embodiment sensations in the 1pp-FBI. These findings suggest that our perception of where our body is in space plays a crucial role in how much we feel it belongs to us, supporting the idea that ownership may be more related to the perceived location than the body itself.

## Introduction

The experience that the body is coherently and continuously part of the self is defined as the sense of body ownership [1, 2]. Previous evidence suggests body ownership results from integrating multisensory signals (i.e., visual, tactile, and proprioceptive) that we constantly receive

**Data Availability Statement:** Data and analysis code are available on the Open Science Framework platform at the following link: https://osf.io/z7nbx/?view_only=92e64fef49a14c9bb96fe8c2b63b96a0.

**Funding:** This work has received funding from the Bial Foundation (https://bialfoundation.com/com/). GT received the award for the project "I am where I believe my body is" (application number: 101/2022; application id: A-37789) in the frame of the Bial Foundation Grants Programme 2022/23 [grant id: G-29862]. The funders had no role in study design, data collection and analysis, decision to publish, or manuscript preparation.

**Competing interests:** The authors have declared that no competing interests exist.

from the environment and the body itself [3]. This continuous integration ensures the persistence and consistency of how we experience our bodies, a fundamental aspect of consciousness and critical for our interaction with the world around us [4].

Experimental manipulations can alter body ownership in healthy participants (i.e., body illusions). For instance, the Rubber Hand Illusion (RHI; [5]) elicits a sense of ownership towards a fake hand through synchronous visuo-tactile stimulation between a fake visible and the real hidden hand. Similarly, in the Full Body Illusion (FBI; [6, 7]), a sense of ownership towards a virtual body is achieved by showing a body frame from either a first-person [8] or a third-person [6] perspective. Curiously, in the RHI, where the fake hand is placed near the real hand (i.e., not spatially aligned), the sense of ownership towards the fake hand is often accompanied by a feeling of disownership toward the real hand [9–11] and a proprioceptive drift toward the fake body [6–8]. In contrast, during the first-person perspective Full Body Illusion (1pp-FBI), where the virtual and real bodies are spatially aligned, participants tend to embody the fake body without disowning the real one [12]. However, when the FBI is administered from a third-person perspective (3pp-FBI), with misalignment between the virtual and real bodies, participants show feelings of disownership towards the real body and a proprioceptive drift toward the fake one again [6–8, 13].

The perspective change in the FBI comes together with the alignment or misalignment of the real and fake bodies, thus suggesting a strict relationship between the sense of body ownership/disownership and the perception of the body's spatial position. In the RHI and the 3pp-FBl, disownership feelings and proprioceptive drift would emerge due to a recalibration of the body's spatial position caused by the misalignment between the real and fake bodies, which does not occur in the 1pp-FBI because the two bodies are aligned in space. Recently, our research group proposed that ownership stems from where we perceive our body to be rather than the body itself [12]. This hypothesis suggests that if people expect their body to be in a specific location, a virtual body in that spot will feel like their own. In the context of the RHI, the embodiment of the fake hand arises when people predict their hand to be in the rubber hand's position. The predicted position clashes with the actual location of their real hand, causing proprioceptive drift and contributing to feelings of disownership for the real hand. Conversely, the 1pp-FBI should not trigger disownership because the real and virtual bodies are aligned. Since there is no discrepancy between the predicted and actual body position, no shift in ownership should occur.

Thus, the study's primary goal was to establish the role of predicting the body's location in shaping the emergence of body ownership and disownership. Specifically, we aimed to determine the association between body ownership and its perceived location while directly comparing the virtual RHI (vRHI) and 1pp-FBI. We manipulated the respective positions of the real and the fake bodies within a virtual reality (VR) setup for both illusions. We considered the aligned condition, similar to the classical 1pp-FBI, where the virtual and real bodies were located in the same position, and the misaligned condition, where the two bodies were not spatially aligned, similarly to the classic RHI. Both explicit (i.e., Embodiment Scale; [11]) and implicit (i.e., Skin Conductance Responses; [14, 15]) measures were used to assess ownership and disownership. SCR captures autonomic responses to perceived threats against an embodied fake or virtual body. When the fake or virtual body is perceived as part of one's own, threats to it elicit emotional and defensive reactions similar to threats to the real body [16]. Previous studies showed that threatening an embodied fake or virtual hand increases SCR significantly, indicating heightened emotional arousal [17, 18]. Furthermore, we conducted a Body Localisation Task to evaluate the body's spatial location prediction.

We expected the following results: (i) increased embodiment sensations towards the virtual body parts after both illusions in all conditions, (ii) higher disownership feelings over the real

body in the misaligned condition after both illusions, (iii) a shift in the perceived body's position (i.e., proprioceptive drift) in the misaligned condition after both illusions. Moreover, we predicted a correlation between embodiment and the perceived body's position: the more participants perceived their body close to the virtual body and with higher precision, the higher the embodiment feelings over the avatar. Similarly, we expected a correlation between disembodiment and the perceived body's position: the more participants perceived their body close to the avatar and with higher precision, the higher the disembodiment feelings in the misaligned conditions but the lower in the aligned conditions.

## Methods

The current project, including hypotheses, design, sampling and analysis plan, was preregistered before the research was conducted at the following link: https://doi.org/10.17605/OSF.IO/XQJPG.

### Participants

Eighty-one healthy subjects participated in the study (49 female, mean age: 22.21 ± 3.09 years; mean school age: 13.86 ± 1.69 years). Recruitment started on September 6, 2023 and ended on November 11, 2023. One participant was excluded due to technical issues. All participants had either normal vision or vision corrected to normal and were unaware of the experiment's aims. Prior to participation, all participants provided their written informed consent. The study received approval from the local Ethics Committee "Commissione per la Valutazione della Ricerca, Dipartimento di Psicologia" at the University of Milano-Bicocca (protocol number: RM-2022-574), and it adhered to the ethical principles of the Declaration of Helsinki [19]. The general purpose of the study and the procedural details were described to participants before their informed consent was obtained. After the experimental procedure, the study's objectives were explained to participants in detail. The sample size was determined to satisfy the expected two main results: a) find a significant correlation, compatible in terms of magnitude with that estimated in a meta-analytic study on the correlation between ownership and proprioceptive drift in the RHI [20]; b) detect a mean difference in a bodily illusion paradigm with within-subject experimental design. A previous meta-analysis [20] revealed that a sample size between 46 and 69 participants is necessary to effectively investigate correlation effects within the embodiment phenomena. Moreover, a priori sample size was also computed for a within-subject ANOVA design using G*power to test the embodiment effect in a bodily illusion paradigm. Fixing Power at 0.80, alpha at 0.05 and effect size at f = 0.65 ($\eta^2_p$ = 0.30 [20, 21]), the analysis indicated that it is necessary a sample of 7 participants. Considering the power analysis results and a potential overestimation of meta-analytic results, a sample size of 80 participants has been planned.

### Experimental design

**Procedure.** Participants were exposed to vRHI and 1pp-FBI in a single experimental session, counterbalancing the order across participants (Fig 1A). Each illusion was presented in two possible location conditions (Fig 1B). In the aligned condition, the avatar was aligned with the real body, while in the misaligned condition, the avatar's left hand or legs were shifted towards the body's midline. Specifically, while a lateral shift to the right can be easily implemented for the hand manipulation, a lateral shift of the leg position requires a change in legs' posture (i.e., transitioning from open to joint legs). The order of the two location conditions was counterbalanced across participants within the same illusion type. In each illusion and location condition, participants observed the virtual body for 60 seconds (i.e., visual exposure)

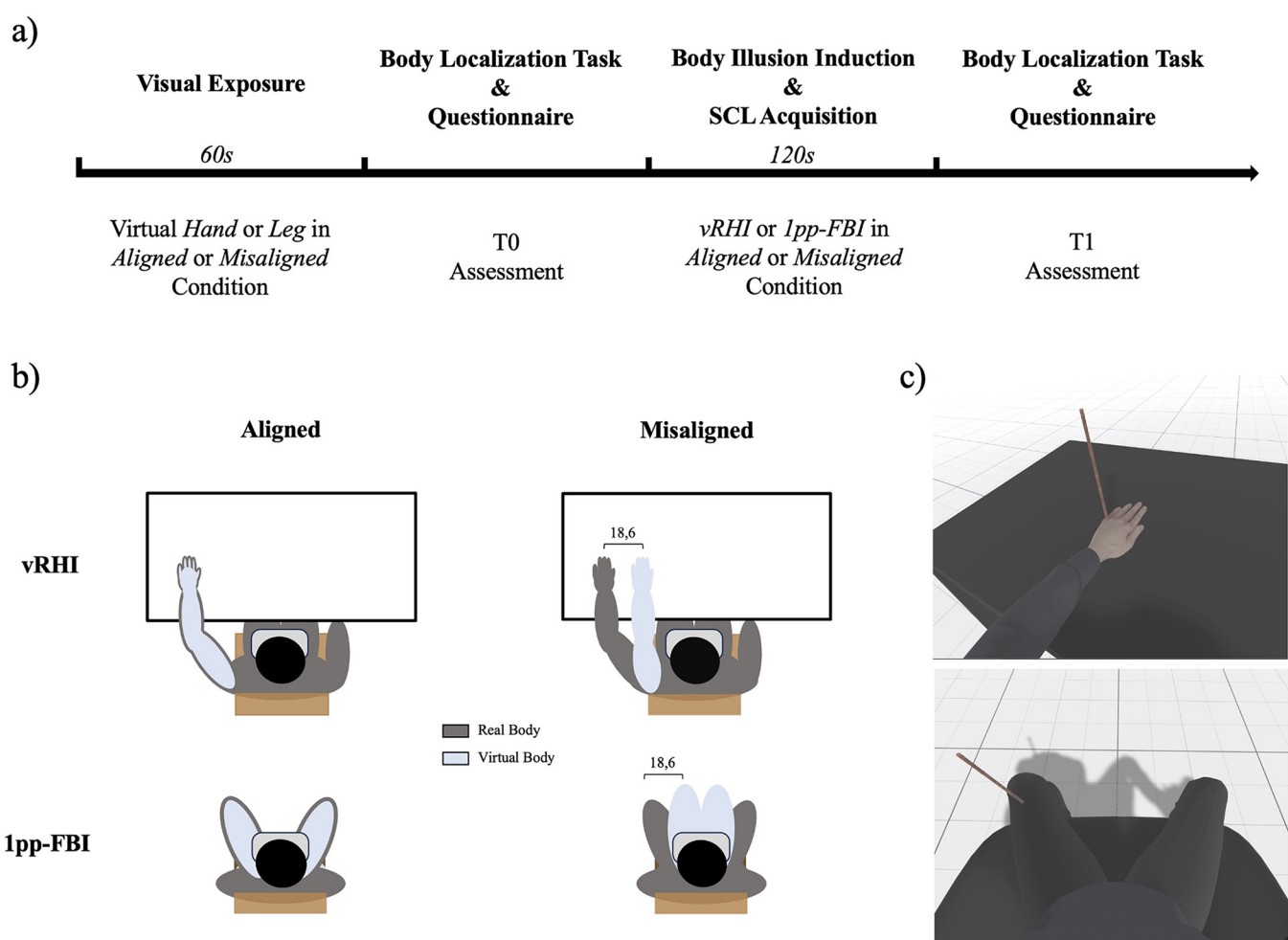

**Fig 1. Experimental procedure and design.** *a) Experimental design.* Participants first observed the virtual body for 60 seconds (i.e., visual exposure) and then experienced the multisensory body illusion. After two minutes of illusion induction, a virtual knife appeared and stabbed the fake virtual body. The skin conductance level (SCL) was collected. After each visual exposure (i.e., T0) and body illusion (i.e., T1), participants performed a Body Localization Task and completed the embodiment questionnaire. *b) Body Illusion type and Location conditions.* Each illusion (vRHI or 1pp-FBI) was presented in two possible location conditions: aligned (i.e., the avatar was aligned with the real body) and misaligned (i.e., the avatar's left hand or legs were shifted towards the body's midline). *c) Body illusion procedure.* Frames extracted from the VR environment depicting the visual-tactile stimulation for the vRHI (upper panel) and 1ppFBI (lower panel). During the illusion, participants received tactile stimulation on their left hand or leg while observing the avatar being synchronously touched in the corresponding area. SCL = Skin Conductance Level; vRHI = virtual Rubber Hand Illusion; 1pp-FBI = first-person perspective Full Body Illusion.

and then experienced the multisensory body illusion. During the illusion, participants received tactile stimulation on their left hand (vRHI) or leg (1pp-FBI) while observing the avatar being synchronously touched in a corresponding area (Fig 1C). After two minutes of visuo-tactile stimulation, a virtual knife appeared and stabbed the fake virtual body. The skin conductance level (SCL) was collected throughout the procedure to capture and monitor the sympathetic response to the virtual threat, according to the illusion and location condition. The stabbing knife served to elicit physiological and emotional responses directly linked to the participant's sense of ownership or disownership. The threatening stimulus (i.e., the stabbing knife) was chosen based on previous literature, proven to evoke a clear and measurable SCL, an indicator of emotional arousal and physiological reactivity [22–24].

After each visual exposure (i.e., T0) and body illusion (i.e., T1), participants performed a Body Localization Task to evaluate the perceived position of the participants' left hand or leg

and filled out the embodiment questionnaire (adapted from [11]; see Supporting Information, S1 Appendix) to measure embodiment and disembodiment subjective experience. Fig 1A shows the procedure timeline. The experiment followed a 2(Time)×2(Illusion)×2(Location) factorial within-subject design, delineating four distinct experimental conditions (vRHI aligned, vRHI misaligned, 1pp-FBI aligned, and 1pp-FBI misaligned) across two times (T0 and T1).

**Apparatus.** Participants were exposed to either the vRHI or the 1pp-FBI. During the vRHI, participants were seated in a chair facing a table, with their left hand located palm down on the table, following specific markers for placing the body midline and the left hand (ensuring a distance from the torso's midpoint to the hand equal to 24 cm). The right hand, equipped with SCL recording electrodes, was positioned under the table on the participant's leg, with the palm facing upward. After being correctly positioned, participants wore the Head Mounted Display (HMD; Oculus Quest 2) in which the virtual environment was presented through Unity 2022 Software by connecting the HMD to the computer (OMEN 30L Desktop GT13-0xxx, Intel Core i9-10900K, 64GB RAM, NVIDIA GeForce RTX 3090). The virtual environment consisted of a neutral space with an avatar sitting on a chair before a table. We selected a white male avatar wearing jeans and a black sweater. We substituted the avatar head with the camera so that the participant's point of view corresponded to the avatar's eyes. In the aligned condition, the virtual hand was positioned identically to the real one (i.e., -33.1 Unity centimetres to the left of the HMD midline). In the misaligned condition, the virtual hand was shifted 18.6 Unity centimetres to the right (i.e., 14.51 Unity centimetres to the left of the HMD midline). The shift in the virtual body's position corresponded to an estimated shift of 14 cm in the natural environment. Fig 1B shows the apparatus.

In the 1pp-FBI, participants sat on a chair positioned away from the table, with their legs spread apart to ensure that the leg opening distance from the torso's midline was 24 cm. In the aligned condition, they observed a virtual body seated on a chair with legs spread apart (i.e., the left and right knees were 27.59 Unity centimetres away from the HMD midline). In the misaligned condition, the virtual legs were close together (i.e., the left and right knees were located 8.99 Unity centimetres away from the HMD midline).

**Body illusions.** During the illusion, participants observed a virtual stick touching the virtual left hand or leg while receiving tactile stimulation at the corresponding location on their hand or leg. Tactile stimuli were delivered through a stick attached to the left controller. The virtual stick's movements were mapped to the HMD left controller so the experimenter could control them. The visuo-tactile stimulation lasted 120 seconds at a frequency of 1 Hz. Upon completion of the multisensory stimulation, a virtual knife appeared and stabbed the virtual left hand or leg. Specifically, after exactly 120 seconds, the experimenter pressed a button to trigger the appearance of the knife. The timing of the knife presentation was carefully coordinated to occur immediately after the visuo-tactile stimulation ended, ensuring that participants remained focused on the virtual body without shifting their attention elsewhere. The knife stabbed the virtual hand in the back, while the leg was stabbed in the thigh. The site of the stabbing corresponded to the prior location of the stick during the visuo-tactile stimulation. The total duration of the appearance and stabbing was approximately 1.5 seconds.

**SCL apparatus and data pre-processing.** SCL was recorded utilising the module GSR100c of a Biopac System MP150 (Goleta, USA). The gain parameter was configured to 5 mmho/V, and the signal was recorded at a sampling frequency of 100 Hz. Two electrodermal activity transducers (SS3LA) were placed on the first phalanx of the middle and ring fingers of the right hand to record the signal. A conductive paste with saline solution was applied to the electrodes to enhance the signal-to-noise ratio. Triggers indicating the onset of the stimulus

were manually sent to the SCL trace using the computer keyboard when the virtual knife appeared to the participants.

The signal was smoothed offline at the end of the recording. The smoothing factor was 25, equivalent to one-fourth of the sampling rate. A digital high-pass filter was applied offline at 0.05 HZ to extract phasic Skin Conductance Response (SCR) [14, 15] from the skin conductance level. The peak-to-peak (P-P) measure [12, 22] evoked by the knife stabbing was taken as an indicator of autonomous nervous system responses and computed for each condition within a time window of 13 seconds, starting from the appearance of the virtual knife. The time window was defined considering the time between the knife's appearance and the stabbing (around 1.5 seconds), the time needed for the SCR to start after the stimulus (estimated in one to three seconds), reach its peak (up to five seconds), and exhaust the response [25].

The SCR P-P following the virtual threat is used as an indicator of the degree of embodiment: higher levels of SCR are typically associated with stronger feelings of ownership over the virtual body, as the virtual body is perceived as part of one's body and threats to it would elicit emotional reactions similar to those triggered by threats to one's body [16, 17]. Conversely, attenuated responses of SCR P-P would indicate a greater sense of disownership towards one's own body [13, 26]. This approach enabled us to quantify physiological reactivity in response to the threat.

**Body localization task.** Before and after the induction of the body illusion, the virtual hand and legs were hidden by a black virtual plane, and participants performed the Body Localization Task in the virtual environment. They were asked to point at the perceived location of the tip of their real left index finger (vRHI) or knee (1pp-FBI) using a virtual coloured ray-cast controlled by the right-hand controller. Specifically, participants were required to perform a ballistic movement from the resting position of their right hand towards their left hand/leg and to point the virtual ray-cast toward the perceived location of the hand/leg as accurately as possible. This measure was repeated ten times to have an implicit estimate of the precision regarding the perceived location (i.e., we calculated the standard error). The Body Location Task was conducted before and after the vRHI and 1pp-FBI in both the aligned and misaligned conditions, resulting in 80 trials.

Two indices were computed from this measure. First, the classical Proprioceptive Drift represents the deviation of the position estimate from the actual hand/leg position. A negative value of the Proprioceptive Drift means that the perceived hand/leg position is shifted to the right relative to their actual position. Conversely, a positive value indicates that the perceived hand/leg position is shifted to the left than the real body-part position. Second, we computed an additional measure (i.e., Virtual Drift) considering the shift from the avatar's position and the prediction's precision (i.e., adding the standard error). Lower scores in the Virtual Drift indicate a greater shift towards the virtual body or higher precision in the estimation. Given our prediction that embodiment depends on the body's location prediction, we wanted to consider if participants predicted their body-part to be where the virtual body-part was and their confidence in that prediction. In a Bayesian framework, it is essential to consider the degree of confidence in the data since it influences the results as much as the data themselves. Compared with the classical proprioceptive drift, this new score will provide information about the drift towards the avatar and the estimate's precision.

Moreover, after performing the Body Localization Task, participants were asked to express their confidence in the indicated location, providing an explicit judgment on the estimate's precision in a 7-point Likert scale.

**Embodiment questionnaire.** Before and after each condition, we also administered the Embodiment Questionnaire to assess feelings of ownership towards the virtual body and disownership towards the real body (adapted from [11]). Participants rated their agreement on 18

questions on a seven-point Likert scale (-3 to +3). Ten questions aimed to capture the embodiment experience, six focused on disembodiment, and two addressed physical sensations. We averaged participants' responses to each dimension to analyse the emergence of the embodiment and disembodiment across conditions (refer to Supporting Information for the questionnaire details, S1 Appendix).

## Data analysis

Analyses were conducted with R 4.3.2 [27]. We conducted a series of Bayesian regression models in a factorial within-subject design since Bayesian analysis can provide evidence supporting both the null and alternative hypotheses. The Bayesian analyses were performed using the *brms* package [28]. We initially assessed the presence of outliers and the distribution type for each dependent variable to define the distribution family. We applied weakly informative priors centred around 0, aligning with those commonly employed in the JASP Software for ANOVA analysis to enhance replicability and reproducibility [29, 30]. We contrast-coded all factors as [-0.5; 0.5] so that the intercept corresponds to the grand mean [31]. After conducting the posterior predictive check, we interpreted results referring to the 95% Credible Intervals (CIs). Moreover, to assess the predicted effects directly, we computed the Bayes Factor to quantify evidence for the complete model relative to a model without the predicted effect. Refer to Supporting Information for additional details regarding the selected priors and posterior predictive check (S2 Appendix).

**Embodiment questionnaire.** We ran a within-subjects 2×2×2 Bayesian regression on the averaged responses for the embodiment and disembodiment factors to analyse the subjective (explicit) aspect of embodiment and disembodiment sensations. Illusion (vRHI vs 1pp-FBI), Location (Aligned vs Misaligned), and Time (T0 vs T1) and their interactions were considered fixed factors. We included participants as a cluster variable to consider inter-subject variability and estimated random intercepts and slopes for the main effects of Illusion, Location and Time. Results were interpreted by inspecting 95% CIs.

**Proprioceptive drift.** After removing improbable responses (< -100 or > 0), we checked the presence of outliers in each subject and condition using the boxplot method implemented in the R function *identify_outliers* (*rstatix* package). Sixty-four participants showed at least one outlier pointing in at least one condition. No one presented more than five (50%) outliers in the same condition, so we removed 293 outliers (i.e., 6.10% of the total sample). Then, the Proprioceptive Drift was calculated by subtracting the estimated position from the real position of the body. A negative value of the Proprioceptive Drift means that the body-part is perceived shifted to the right relative to the actual position. Conversely, a positive value indicates that the body-part is perceived as shifted to the left. We performed a 2×2×2 Bayesian regression on this index to assess the illusion influence on the body's location prediction. Illusion, Location, and Time and their interactions were considered fixed factors. We included participants and trials as cluster variables to consider inter-subject and inter-trial variability. We estimated random intercepts and slopes for the main effects of Illusion, Location and Time. Results were interpreted by inspecting 95% CIs.

**Virtual drift.** Pointing estimates were also used to calculate the Virtual Drift by subtracting the estimated position from the avatar's position. Moreover, to obtain the Virtual Drift composite score, we averaged the absolute value of all responses from each participant in each condition (i.e., 10 pointing) and added the standard error. Lower scores in the Virtual Drift indicated a greater shift towards the virtual body or higher precision in the estimation and vice-versa.

To investigate the relation between the prediction of the body's location and embodiment feelings, we computed correlations between the Virtual Drift and the subjective ratings from

the questionnaire at T1. For this purpose, we implemented a model with Location and Illusion as fixed effects for each variable (i.e., embodiment and virtual drift). We included participants as a cluster variable and estimated random intercepts and slopes for the main effects of Illusion and Location. Then, we extracted participants' random intercepts as an index of score variability. Indeed, random intercepts represent each participant's expected mean considering all conditions. We correlated the random intercepts extracted from models. This method allows us to measure individual differences independently by the other effects.

Given the formulation of two distinct hypotheses regarding the correlation between virtual drift and disembodiment in aligned and misaligned locations, we extracted random intercepts from two separate models in T1. We considered aligned and misaligned conditions separately. We implemented models with Illusion as a fixed effect for each variable (i.e., disembodiment and virtual drift), including participants as a cluster variable and estimating random intercepts and slopes for the main effect of Illusion. Then, we extracted participants' random intercepts and correlated disembodiment and virtual drift scores separately for aligned and misaligned.

We also performed the correlation between implicit (i.e., virtual drift standard error) and explicit (i.e., a 7-point Likert scale answer) judgments of the estimate's precision by extracting the random intercepts again. We implemented a model with Time, Location and Illusion as fixed effects for each variable (i.e., virtual drift standard error and 7-point Likert scale answer). We included participants as a cluster variable and estimated random intercepts and slopes for all main effects. Then, we correlate participants' random intercepts.

**SCR.**   The SCR was considered to examine the physiological response to the embodiment phenomena. Thus, a 2 (Illusion) ×2 (Location) Bayesian regression design was applied to the SCR P-P index. Illusion, Location, and their interactions were considered as fixed factors. We included participants as a cluster variable to consider inter-subject variability and estimated random intercepts and slopes for all main effects. Results were interpreted by inspecting 95% CIs. Also, a correlation between the SCR P-P index and the subjective ratings of embodiment and disembodiment responses was performed. For this correlation, we extracted participants' random intercepts from the SCR model and participants' random slope of Time from the models assessing embodiment and disembodiment. Thus, we computed the correlation between participants' SCR variability at T1 and the individual variation of the embodiment and disembodiment difference between T0 and T1.

Data and analysis code are available on the Open Science Framework platform at the following link: https://osf.io/z7nbx/?view_only=92e64fef49a14c9bb96fe8c2b63b96a0

## Results

### Embodiment questionnaire

The Bayesian regression conducted on embodiment scores indicates a three-way interaction between Time, Illusion, and Location. Specifically, the difference between T0 and T1 is higher in the misaligned than in the aligned condition when considering the 1pp-FBI rather than the vRHI (Time×Location×Illusion: 0.38; 95% CI: [0.03; 0.73]). Results also show all main effects and two-way interactions (Table 1). As shown in Fig 2A, in the aligned condition, embodiment levels increased after both illusions (T1), but high embodiment feelings were already present before the illusions, following the visual exposure (T0). Similarly, embodiment increased after both illusions (T1) in the misaligned conditions. This difference was more pronounced following the 1pp-FBI, compared with the vRHI, which exhibited a high level of embodiment even before the illusion (T0). To address our first hypothesis of increased embodiment sensations after the illusions in all conditions, we computed the Bayes Factor (BF) for Time effect

**Table 1. Results of the Bayesian regression ran on the embodiment responses.**

| Embodiment Effects | Estimate | Est. Error | Lower 95% CI | Upper 95% CI |
|---|---|---|---|---|
| *Intercept* | 0.73 | 0.12 | 0.49 | 0.96 |
| $Time_{T1}$ | 0.48 | 0.06 | 0.37 | 0.59 |
| $Location_{misaligned}$ | -0.73 | 0.09 | -0.91 | -0.54 |
| $Illusion_{1pp\text{-}FBI}$ | -0.39 | 0.10 | -0.59 | -0.20 |
| $Time_{T1}*Location_{misaligned}$ | 0.52 | 0.09 | 0.33 | 0.70 |
| $Time_{T1}*Illusion_{1pp\text{-}FBI}$ | 0.33 | 0.09 | 0.14 | 0.51 |
| $Location_{misaligned}*Illusion_{1pp\text{-}FBI}$ | -1.00 | 0.10 | -1.20 | -0.81 |
| $Time_{T1}*Location_{misaligned}*Illusion_{1pp\text{-}FBI}$ | 0.38 | 0.18 | 0.03 | 0.73 |

The table shows the mean (Estimate) and the standard deviation (Est. Error) of the posterior distribution of each effect with the 95% Credible Intervals (lower 95% CI, upper 95% CI). In bold, the posterior distributions without a zero overlapping.

inclusion by comparing the full model with a model without the Time effect. The model comparison reveals that data are better explained when including Time as a factor (BF = 1.073x10²⁰).

S3 Appendix reports the results for the physical sensations factor, which is part of the questionnaire but not the main focus of the present study.

Considering the results of the Bayesian regression on the disembodiment score, the difference between T0 and T1 is higher in the 1pp-FBI condition than in the vRHI (Time×Illusion: 0.19; 95% CI: [0.03; 0.35]). Moreover, the location difference is higher in the 1pp-FBI than in the vRHI (Location×Illusion: 0.26; 95% CI: [0.09; 0.42]). Indeed, the sense of disembodiment is higher after the 1pp-FBI compared to the vRHI, irrespective of the spatial location of the virtual body. Moreover, disembodiment is stronger in the misaligned than aligned condition in the 1pp-FBI, regardless of the assessment time. Results also show main effects of Time and Location (Table 2), but, contrary to our expectation, no changes in disownership feeling seem

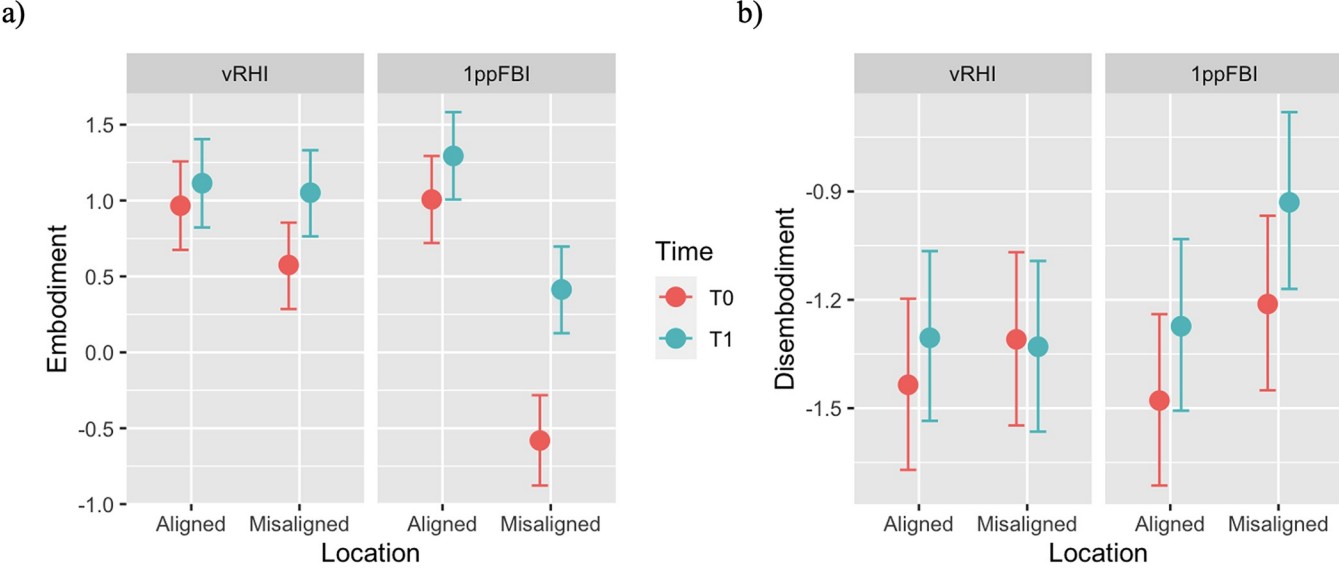

**Fig 2. Questionnaire results.** The figure shows the results of the Bayesian regression on the averaged answers to Embodiment (a) and Disembodiment (b) statements. Error bars indicate 95% Credible Interval limits.

**Table 2. Results of the Bayesian regression ran on the disembodiment responses.**

| Embodiment Effects | Estimate | Est. Error | Lower 95% CI | Upper 95% CI |
|---|---|---|---|---|
| *Intercept* | **-1.28** | **0.10** | **-1.47** | **-1.08** |
| $Time_{T1}$ | **0.15** | **0.05** | **0.06** | **0.25** |
| $Location_{misaligned}$ | **0.18** | **0.06** | **0.06** | **0.29** |
| $Illusion_{1pp-FBI}$ | 0.12 | 0.08 | -0.02 | 0.28 |
| $Time_{T1}*Location_{misaligned}$ | -0.04 | 0.08 | -0.20 | 0.12 |
| $Time_{T1}* Illusion_{1pp-FBI}$ | **0.19** | **0.08** | **0.03** | **0.35** |
| $Location_{misaligned} * Illusion_{1pp-FBI}$ | **0.26** | **0.08** | **0.09** | **0.42** |
| $Time_{T1}*Location_{misaligned}Illusion_{1pp-FBI}$ | 0.23 | 0.16 | -0.08 | 0.54 |

The table shows the mean (Estimate) and the standard deviation (Est.Error) of the posterior distribution of each effect with the 95% Credible Intervals (lower 95% CI, upper 95% CI). In bold, the posterior distributions without a zero overlapping.

to occur in the vRHI (Fig 2B). Finally, despite the observed differences between conditions in terms of disembodiment, the questionnaire scores were consistently negative, indicating that participants did not explicitly report remarkable disownership feelings.

We also computed the BF to test evidence in favour of including the expected effect (i.e., the interaction between Time and Location; see hypothesis 2). We thus compared the full model with a model without the interaction effect Time×Location. The model comparison reveals that data are better explained when the interaction Time×Location is not included (BF = 0.00041).

## Proprioceptive drift

The Bayesian regression on the Proprioceptive Drift revealed that the difference between T0 and T1 is higher in misaligned than in aligned condition (Time×Location: -2.49; 95% CI: [-2.98; -2.00]), regardless of the type of illusion. This result suggests a shift of the perceived position towards the right (i.e., towards the virtual body position) after both illusions in the misaligned condition. As expected, after both illusions, a recalibration of the position of the hand/leg in space toward the shifted virtual body is observed. In contrast, no changes seem to emerge in the aligned condition after the two illusions (Fig 3). Also, the difference between pre and post is smaller in the 1pp-FBI than in the vRHI (Time×Illusion: -0.60; 95% CI: [-1.08; -0.12]), suggesting that the perceived hand position is more influenced by the visuo-tactile stimulation than the perceived leg position. Moreover, a leftward bias emerged in estimating the hand/leg position, as if the body-part were perceived further from the body midline. As hypothesised (see hypothesis 3), the BF for Time×Location interaction inclusion, computed by comparing the full model with a model without the interaction effect, revealed that data are better explained when the Time×Location interaction is included (BF = $2.293 \times 10^{22}$). See Table 3 for detailed results.

## Virtual drift

We found anecdotal to moderate evidence in favour of the absence of a correlation between Virtual Drift and both Embodiment (r = -0.111 $BF_{10}$ = 0.402) and Disembodiment (Aligned condition: r = 0.087, $BF_{10}$ = 0.336; Misaligned condition: r = 0.058 $BF_{10}$ = 0.289). These results suggest that embodiment and disembodiment do not vary with the prediction of the body's location in space.

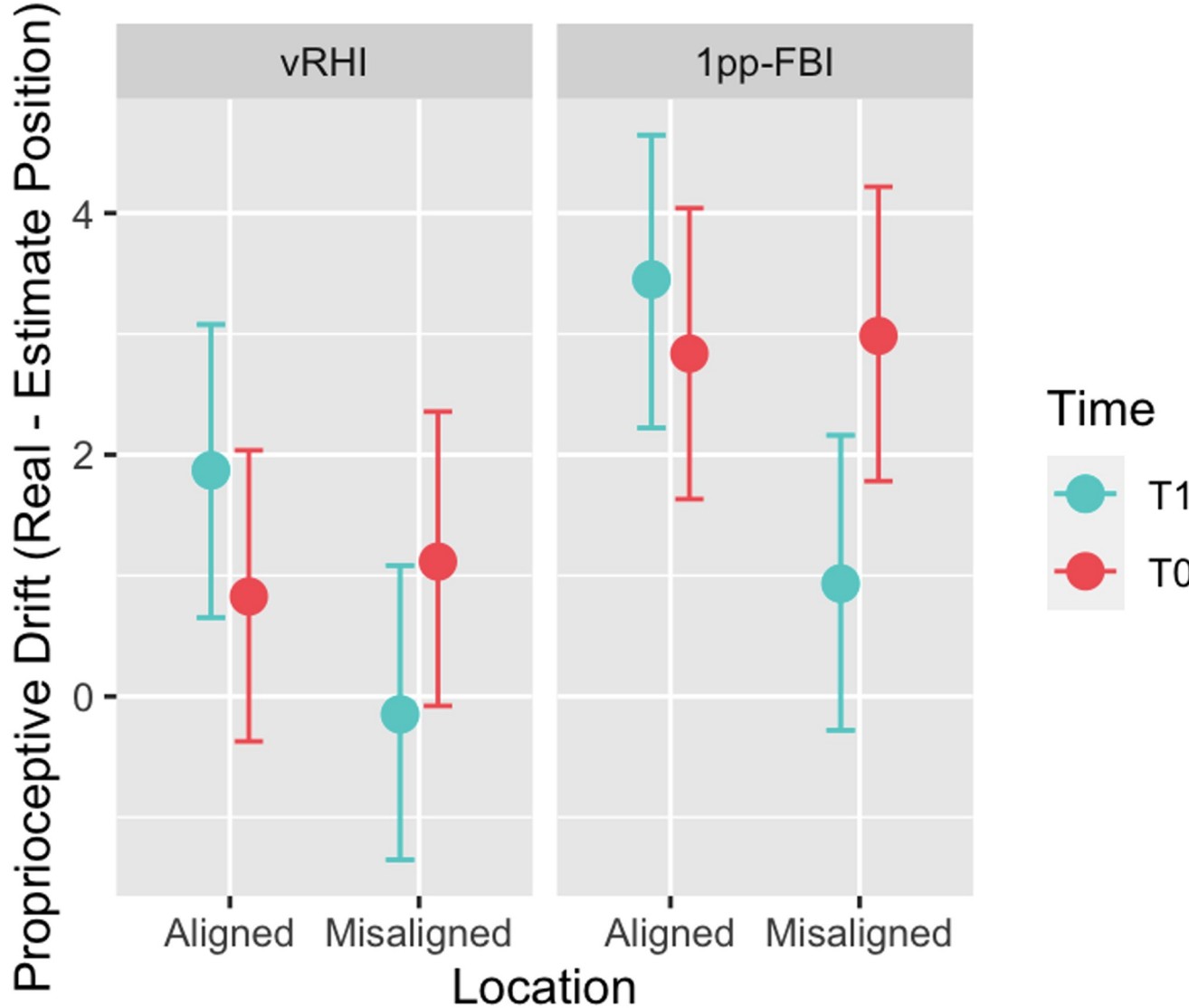

**Fig 3. Results of Bayesian regression on the proprioceptive drift.** The figure shows the results of the Bayesian regression on the Proprioceptive Drift (Real–Estimate position), depending on Illusion, Time, and Location; Error bars indicate 95% Credible Interval limits.

Similarly, the correlation between implicit (i.e., virtual drift standard error) and explicit (i.e., a 7-point Likert scale answer) judgments of estimate precision revealed moderate evidence in favour of the absence of a correlation ($r = 0.027$; $BF_{10} = 0.262$).

However, the strength of the evidence suggests taking these conclusions cautiously.

### SCR

The Bayesian regression suggested that the SCR P-P is lower in the 1pp-FBI than in the vRHI (Illusion: -0.20, 95% CI [-0.37; -0.04]). See S4 Appendix in the Supporting Information for the detailed table results and results plot. The correlation between SCR P-P and subjective ratings of embodiment and disembodiment revealed anecdotal evidence in favour of the absence of correlations (Embodiment: $r = 0.1322$, $BF_{10} = 0.480$; Disembodiment: $r = 0.142$, $BF_{10} = 0.524$).

**Table 3. Results of the Bayesian regression ran on the proprioceptive drift.**

| Embodiment Effects | Estimate | Est. Error | Lower 95% CI | Upper 95% CI |
|---|---|---|---|---|
| *Intercept* | **1.74** | **0.46** | **0.82** | **2.65** |
| *$Time_{T1}$* | -0.42 | 0.29 | -0.98 | 0.15 |
| *$Location_{misaligned}$* | **-1.03** | **0.52** | **-2.04** | **-0.01** |
| *$Illusion_{1pp-FBI}$* | **1.61** | **0.47** | **0.68** | **2.53** |
| *$Time_{T1}$\*$Location_{misaligned}$* | **-2.49** | **0.25** | **-2.98** | **-2.00** |
| *$Time_{T1}$\* $Illusion_{1pp-FBI}$* | **-0.60** | **0.24** | **-1.08** | **-0.12** |
| *$Location_{misaligned}$ \* $Illusion_{1pp-FBI}$* | -0.32 | 0.25 | -0.79 | 0.17 |
| *$Time_{T1}$\*$Location_{misaligned}Illusion_{1pp-FBI}$* | -0.36 | 0.38 | -1.09 | 0.38 |

The table shows the mean (Estimate) and the standard deviation (Est.Error) of the posterior distribution of each effect with the 95% Credible Intervals (lower 95% CI, upper 95% CI). In bold, the posterior distributions without a zero overlapping

## Additional analysis and results

During the experimental sessions, we noticed a potential bias induced by the order of experimental conditions (Aligned vs. Misaligned presented as the first condition) in the vRHI. Participants' perception of hand position seemed influenced by the first condition experienced. We thus decided to run exploratory analyses, in addition to the preregistered ones, to investigate this specific aspect.

**Influence of the Order of Sessions on Question 6.** Question 6 of the embodiment questionnaire (i.e., "*It seemed like my hand was in the location where the virtual hand was*") reflects the explicit perception of the hand's position relative to the virtual one. We initially focused on the vRHI, conducting an additional Bayesian regression on Question 6 because the abovementioned effect was observed for the vRHI. We considered Time and Location as within-subjects fixed factors, and the Order of Sessions was set as a between-subjects fixed factor. We considered participants as a cluster variable and calculated random slope and intercept for the main effects of time and location.

The results showed that the difference between aligned and misaligned locations was stronger when subjects first performed the aligned condition rather than the misaligned (4.32; 95% CI: [2.41; 6.46]). As shown in Fig 4A, participants perceived the real hand to be in a similar position as the virtual hand in both conditions (i.e., aligned and misaligned) when they underwent the misaligned condition first. Conversely, when the aligned location was the initial condition, participants correctly perceived the difference in the two positions (Order of Sessions×Location: 0.24; 95% CI: [-2.53; 3.05]).

We then applied the same model to the 1pp-FBI. Interestingly, in this case, the order in which the sessions were presented did not affect the perception of the virtual leg position. See S5 Appendix in the Supporting Information for detailed table results.

**Influence of the order of sessions on proprioceptive drift.** We also tested whether the order of session presentation influenced the implicit perception of the hand position in space (i.e., Proprioceptive Drift) during the vRHI. Thus, we computed the same Bayesian regression on Proprioceptive Drift in the vRHI. However, no evidence emerged for the interaction effect between Location and Order of Session (95% CI: [-1.14; 0.73]), suggesting that participants' estimates are not influenced by the order of presentation of conditions (Fig 4B). See S5 Appendix in the Supporting Information for detailed table results.

**Correlations only considering the 1pp-FBI.** Considering this potential confounding factor in the vRHI, we examined the correlation between Virtual Drift and the subjective ratings

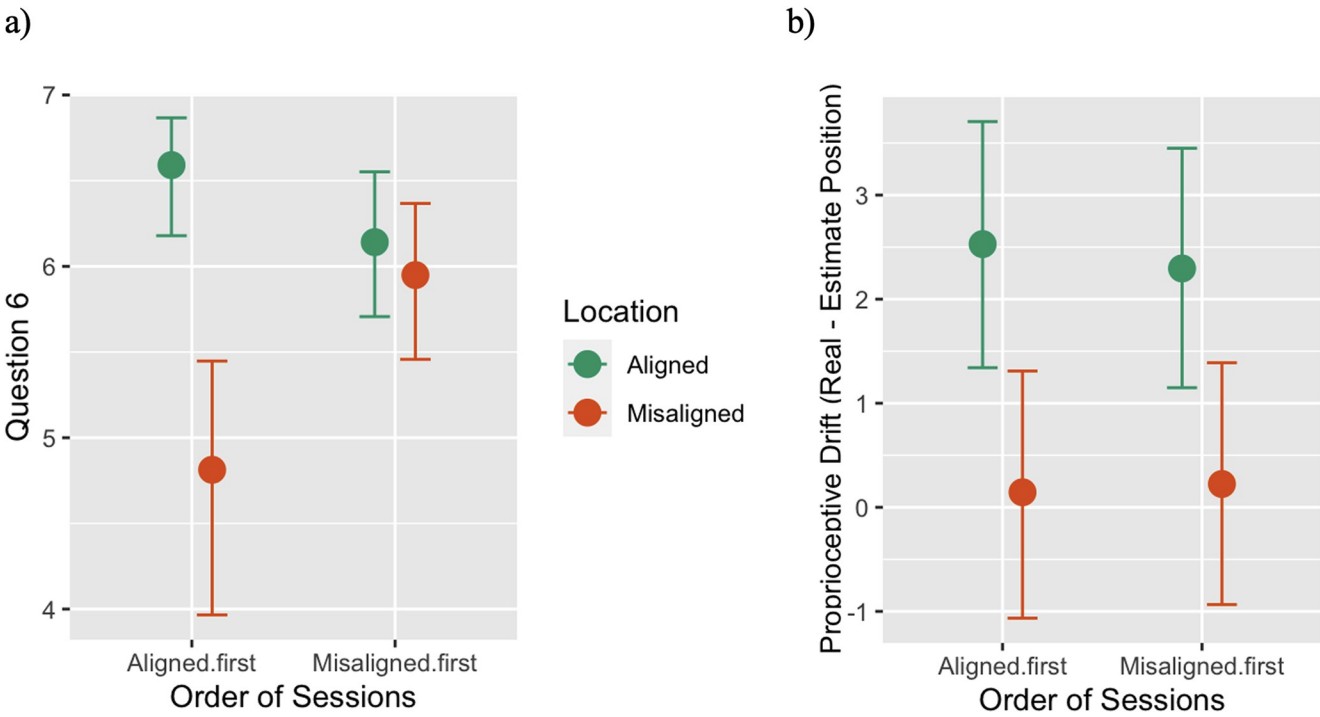

**Fig 4. Results of additional Bayesian regression for the vRHI.** a) Results of the Bayesian regression on Question 6 depending on the Order of Sessions, Time, and Location; b) Results of the Bayesian regression on Proprioceptive Drift depending on the Order of Sessions, Time, and Location; Error bars indicate 95% Credible Interval limits. Given that the cumulative distribution provided the most accurate fit for the data, we recode the variable. We transformed the original scale from 1 to 7, thus creating an integer variable without zero.

of Embodiment and Disembodiment across each location condition only in the 1pp-FBI conditions (i.e., the ones unbiased by the order of administration). For Embodiment, we found anecdotal evidence in favour of the presence of a correlation in the misaligned condition ($r = -0.224$, $BF_{10} = 1.657$) and in favour of the absence of the correlation in the aligned condition ($r = -0.107$, $BF_{10} = 0.388$). Although the evidence is weak, results suggest a trend in line with our hypothesis about embodiment in the misaligned condition: participants who predicted their left leg to be close to the virtual leg also showed high embodiment sensations. On the contrary, when the virtual and the real leg were placed in the same location (i.e., aligned condition), the body's location prediction did not influence embodiment towards the virtual body. Regarding the Disembodiment, we found moderate evidence in favour of the absence of correlation both in aligned ($r = -0.065$, $BF_{10} = 0.298$) and in misaligned ($r = -0.016$, $BF_{10} = 0.258$) conditions, suggesting that the body's location prediction is not associated with the disembodiment feelings in this experimental procedure.

## Discussion

Body and spatial perceptions are entangled [3, 4]. Here, we hypothesised that ownership may be more related to the perceived location than the body itself and investigated the role of predicted body-part position in shaping ownership. To this aim, we employed two body illusions, the vRHI and the 1pp-FBI, to manipulate the perceived locations of body-parts in space, altering the respective positions of the real and virtual bodies.

The essential illusory contrast was the congruent multisensory stimulation against the visual exposure to the avatar instead of a multisensory incongruent stimulation as most

frequently done [5, 6, 8]. The critical element of novelty was comparing illusory conditions that always induce embodiment but potentially with different strengths. By doing so, we identified a stricter control contrast than typically done when congruent and incongruent visuotactile stimulations are compared. In line with our hypotheses and previous results [11, 13, 21, 32], we found that the sense of embodiment increased after the two illusions, both when the virtual and real bodies were in different spatial locations and when they were co-located. Specifically, the visuo-tactile stimulation during the vRHI induced similar embodiment levels in the aligned and misaligned conditions (see [32] for a similar result). Interestingly, in the aligned condition, ownership was already high after the mere visual exposure (i.e., T0 before visuo-tactile stimulation), suggesting that simply viewing a fake body spatially superimposed to my body can induce ownership toward it [7, 33, 34]. Additionally, for the vRHI, ownership levels were high even in the misaligned condition prior to visuo-tactile stimulation. This aligns with the Self-Avatar Follower Effect [35–37], where an incongruence between real and virtual bodies can initially promote embodiment sensations, though embodiment may diminish if the mismatch persists or increases, potentially leading to a break in embodiment [36, 37]. In our study, ownership remained intact despite the mismatch, possibly due to the dominance of visual cues, with no break in embodiment. Furthermore, the multisensory congruence during the visuo-tactile stimulation further enhanced the sense of embodiment.

It is worth noticing that when it comes to the 1pp-FBI, the difference between aligned and misaligned locations, after the mere visual exposure, increases compared to the vRHI. Specifically, after the visual exposure to the 1pp-FBI, we observed lower levels of embodiment in the misaligned than in the aligned condition. Simple visual exposure is rarely investigated in body illusions [7, 33, 34], and nobody has yet directly compared different illusions on the same participants. Thus, our results highlight an interesting novel aspect. Despite the similarities in the two illusory experimental procedures, results show specific patterns, aligning with previous studies that suggest differences in hand processing compared to full-body ownership [38, 39]. The hand seems more prone to embodiment even in the case of misalignment, possibly because of its unique features, serving as a crucial stimulus for interaction and being specially represented cognitively and neurally [40–42]. Another possibility is that the FBI requires combining proprioceptive information across the entire body. If this is the case, visual information may play a less crucial role, making the initial visual exposure less effective in establishing the illusion. Indeed, the FBI is related to a global body representation processing, involving integrating distributed somatosensory, vestibular, and proprioceptive inputs from multiple body parts. Contrary to the more localised rubber hand illusion (RHI), where proprioceptive and visual signals are limited to the hand, the FBI involves a broader integration of sensory modalities that contribute to the representation of the body as a whole [38, 43, 44].

By directly comparing two illusions, we also found that the SCR to a virtual threat is stronger after the vRHI than after the 1ppFBI, indicating that a threat to the hand induces a greater arousal response than the same threat on the leg. Again, this effect could be due to the hand's unique features and it is in line with the embodiment results, showing lower scores after the 1pp-FBI. Another possible interpretation is that the perceived distance to the threat may influence the observed difference in arousal response. Specifically, a threat to the hand may be perceived as closer and, therefore, more alarming than a threat to the leg, potentially resulting in a stronger SCR. Unfortunately, we did not find a correlation between the SCR-PP and the embodiment or disembodiment sensation, suggesting that different measures (i.e., implicit and explicit) capture different facets of the illusions, as already shown in previous studies [7, 21, 45, 46].

In body illusion, the sense of ownership towards the fake body is often accompanied by a feeling of disownership toward the real one [6–11, 13]. Our results on the 1pp-FBI confirmed

increased disembodiment feelings for the legs after the multisensory illusion and when the virtual legs were dislocated compared to being co-located with the real body. These findings are consistent with our hypothesis and experimental results showing disembodiment after the 3pp-FBI [7, 13] or its absence after the 1pp-FBI [12]. As mentioned earlier, the 3pp-FBI can be associated with our misaligned condition because of the incongruence between the real body position and the virtual body position. Conversely, the 1pp-FBI resembles our aligned condition because of the bodies' co-location.

Disembodiment results from the vRHI go against our initial hypothesis at first sight. It seems that no changes in disownership feelings occurred for the vRHI, which differs from previous findings [9, 11]. While this result surprised us, by debriefing participants at the end of the experimental session, we noticed a potential confound determined by the order of presentation of the conditions. Indeed, only in the vRHI, when the misaligned condition was presented before the aligned one, participants reported alignment between the real and the fake hand in both the aligned and misaligned conditions. On the other hand, if the aligned condition was shown at first, participants adequately discriminate the two spatial conditions (see Fig 4A for this specific investigation). In the case of the 1pp-FBI, participants always correctly discriminate the aligned and misaligned conditions, no matter what condition was shown first. While this aspect opens new research questions, we can hypothesise that in a vRHI setup, where all visual inputs are controlled, and proprioceptive information is limited to a body part, visual capture is so strong to drive this spatial misidentification. Accordingly, if the virtual hand is first shifted from the actual one, a larger space would be available to represent the hand's location (i.e., a larger space of embodiment). Disownership may not change in this context since there is no explicit perception of a mismatch between the virtual and real hand positions. Suppose the failure to recognise the position mismatch is due to strong visual capture; the 1pp-FBI may balance this effect by emphasising proprioceptive information from the whole body. Indeed, a wrong body's position is harder to consider as our own if we focus on the proprioceptive information coming from the body as a whole. Crucially, a lateral shift of the leg position in our manipulation requires a change in leg posture (i.e., open legs vs. joint legs), emphasising the difference between the two leg positions and increasing the weight of proprioceptive information. Nonetheless, multisensory stimulation may overcome proprioceptive information, thus inducing the illusion.

Moreover, it is important to note that distance perception in VR can differ from real-world perception [47–49] with various factors contributing to these discrepancies [50]. Thus, it is possible that the immersive nature of VR and the way the body is represented within the virtual environment may cause participants to perceive the virtual limb's position differently than they would in a real-world setting.

To assess the implicit prediction of the body-part's position, we considered the change in the proprioceptive drift after the illusion (i.e., the difference between the real and estimated position) across conditions. As expected, we found that the perceived hand/leg location shifted toward the virtual body-part after both illusions but only in the misaligned conditions. In aligned condition, the perceived body-part position remained relatively unchanged. These results are consistent with the previous works on the classical RHI and 3pp-FBI in which the perceived position is shifted toward the fake/virtual body position [5, 9, 13]. Moreover, the perceived hand position may be more influenced by the visuo-tactile stimulation than the perceived leg position, as evidenced by a stronger shift observed after the RHI. Again, this result suggests a higher susceptibility to the vRHI than the 1pp-FBI. Interestingly, a shift of the hand's position perception consistent with the avatar's position emerged even when participants performed the misaligned condition first. These results contrast the erroneous explicit recognition of the hand's position and suggest that explicit and implicit measures, once again,

capture different facets of the illusion. Since the vRHI did not induce any changes in disembodiment feelings, it is possible that consciously recognizing the mismatch between the real and virtual body-part position is more important for triggering disownership than having a prediction of the body's location.

To tackle these hypotheses, we explored the association between the parametric prediction of the body's location and the occurrence of both ownership and disownership sensations. To this aim, we computed a novel index which considers the shift from the avatar position and the precision of the estimation. The general analyses did not support the presence of any correlations. However, our hypotheses seem partially supported in a follow-up analysis focused on the 1pp-FBI. The results have little evidence and must be taken cautiously. However, they show a weak correlation between the shift in the perception of body position in space and embodiment feelings in the misaligned condition. This result suggests that when there is a discrepancy between proprioceptive, tactile, and visual information, as observed in cases of the misaligned location, the level of embodiment is linked to the body's perceived position (and its precision). This correlation may not necessarily emerge when there is no conflict, such as in aligned conditions. Therefore, the hypothesis of a connection between body ownership and spatial location seems partially supported in the 1pp-FBI. Indeed, a change in the body's predicted location emerged in the misaligned condition, shifting from the actual body's position to that of the virtual body. Such recalibration may contribute to the emergence of ownership and disownership feelings, supporting the idea that body ownership might be more connected to the perceived spatial location of our body rather than being a property of the body itself. Our findings can be extended to several clinical conditions involving body representation disruptions, such as somatoparaphrenia and phantom limb. If our hypothesis is correct, the symptoms observed in somatoparaphrenic patients (i.e., the perception that a body part does not belong to oneself) may be explained by a failure to correctly update the position of the body part in space which would not be in the predicted location [51, 52]. Similarly, in the case of phantom limb patients who perceive ownership over the amputated limb, this phenomenon can be explained by the persistence of the predicted position of the amputated limb. Considering our results, effective interventions can be developed for a broader range of body representation disturbances, providing significant therapeutic benefits. These interventions can take advantage of virtual reality to effectively recalibrate the perceived body position in space, thereby manipulating body ownership sensations.

## Conclusion

Disownership was modulated only in the 1pp-FBI, where there was always an explicit recognition of the misalignment between the real and virtual body in addition to an implicit position recalibration. Moreover, the perceived body's position correlated with the level of embodiment feelings when a position recalibration was needed. These results suggest that the shift of the predicted body's position may contribute to the sense of ownership and disownership when there is an explicit recognition of the incongruence between the two bodies' positions.

Overall, the results showed that the perceived body location could be associated with the emergence of ownership and disownership feelings, supporting the idea that ownership may be a property of the perceived spatial location of the body.

## Supporting information

**S1 Appendix. Embodiment Scale (ES).**
(DOCX)

**S2 Appendix. Models and posterior predictive check for each analysis performed.**
(DOCX)

**S3 Appendix. Additional embodiment scale results.**
(DOCX)

**S4 Appendix. SCR-PP results.**
(DOCX)

**S5 Appendix. Additional analyses.**
(DOCX)

## Acknowledgments

The University of Milan-Bicocca supported this work. We thank Doctor Luca Pieri, Doctor Tommaso Brega, and the Mind and Behavior Technological Center (MIBTEC) of the University of Milan-Bicocca for their technical support in developing the virtual environment.

## Author Contributions

**Conceptualization:** Daniele Romano, Giorgia Tosi.

**Data curation:** Francesca Frisco, Vito Bruno, Giorgia Tosi.

**Formal analysis:** Francesca Frisco, Vito Bruno, Giorgia Tosi.

**Investigation:** Francesca Frisco, Vito Bruno, Giorgia Tosi.

**Methodology:** Giorgia Tosi.

**Project administration:** Giorgia Tosi.

**Supervision:** Daniele Romano, Giorgia Tosi.

**Visualization:** Francesca Frisco, Giorgia Tosi.

**Writing – original draft:** Francesca Frisco, Vito Bruno, Giorgia Tosi.

**Writing – review & editing:** Francesca Frisco, Daniele Romano, Giorgia Tosi.

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
