## [Decision Letter · Decision Letter 0]

27 Sep 2024

PONE-D-24-08955I am where I believe my body is: the interplay between body spatial prediction and body ownershipPLOS ONE

Dear Dr. Tosi,

Thank you for submitting your manuscript to PLOS ONE. My apologies again for the long time it has taken to get reviews. The reviewers are positive but have some recommendations for a revision. Therefore, we invite you to submit a revised version of the manuscript that addresses the points raised during the review process.

We look forward to receiving your revised manuscript.

Kind regards,

Prof Jane Aspell

Academic Editor

PLOS ONE

Journal Requirements:

1. When submitting your revision, we need you to address these additional requirements. Please ensure that your manuscript meets PLOS ONE's style requirements, including those for file naming. The PLOS ONE style templates can be found at https://journals.plos.org/plosone/s/file?id=wjVg/PLOSOne_formatting_sample_main_body.pdf and https://journals.plos.org/plosone/s/file?id=ba62/PLOSOne_formatting_sample_title_authors_affiliations.pdf

Reviewers' comments:

Reviewer's Responses to Questions

**Comments to the Author**

1. Is the manuscript technically sound, and do the data support the conclusions?

Reviewer #1: Yes

Reviewer #2: Yes

2. Has the statistical analysis been performed appropriately and rigorously? 

Reviewer #1: Yes

Reviewer #2: Yes

3. Have the authors made all data underlying the findings in their manuscript fully available?

Reviewer #1: Yes

Reviewer #2: Yes

4. Is the manuscript presented in an intelligible fashion and written in standard English?

Reviewer #1: Yes

Reviewer #2: Yes

5. Review Comments to the Author

Reviewer #1: Overall, this manuscript is well written, methodologically strong, and scientifically interesting.

The main issue I have with the manuscript is in the possible misinterpretation of the term ‘body location’, which is used here to speak of the location of body parts (hand or legs), while it is probably most often understood as the location of the body in space, i.e. self-location. This leads to possible misinterpretation of the results and of the experimental manipulation. Please ensure to rephrase the manuscript with term that avoids this ambiguity.

e.g. l 224: “toward the perceived body’s location” : this should read “towards the perceived hand location”.

Please ensure this is the case throughout the manuscript (e.g. l 383, 390)

The description of the experimental conditions for experiment 1pp-FBI could benefit from a clarification.

L 135 says “left hand or legs were shifted towards the body's midline” : this describes a condition where there would be a lateral translation of the legs (like for the RHI).

However, the figure 1 shows that the “dislocation” condition is a difference in body posture (joint knees vs. open legs). This is a very important as the difference between conditions of the 1pp-FBI is therefore driven by the proprioception of legs postures (not a judgement of self-location in space).

L494 “ suggesting that simply viewing a fake body spatially superimposed to my body can induce ownership toward it” This is a very interesting observation that could be described more and related to the concept of ‘break in embodiment’ described in the literature.

The SCR data show a difference for a threat to the hand vs. leg (l509): it is however not discussed that this difference could be due to the perceived distance to the threat (threat to the hand being closer to the view than threat to the leg).

Also, related to the discussion on the perception of limb location (l533), an interpretation that could be discussed is the difference in distance perception in VR compared to reality (often reported as the difficulty to estimate distance, but in general to a distortion of distance perception in VR compared to reality).

Please ensure the English terminology is appropriate: e.g. ‘dislocation’ of an arm would rather be a medical injury with a dislocated joint.

Reviewer #2: This study investigates how our ability to predict our body's spatial position influences feelings of ownership and disownership using two illusion techniques: the vRHI and the first-person perspective 1pp-FBI. Participants viewed a virtual body aligned or misaligned with their own, followed by synchronous tactile stimulation and a stabbing event. SCR was measured as an implicit embodiment indicator, along with a Body Localization Task and a questionnaire. Results showed that both illusions increased ownership, but this was weaker in the misaligned 1pp-FBI. Disownership occurred only in the misaligned 1pp-FBI, especially when legs were misaligned. Additionally, participants recalibrated their perceived body position toward the virtual misaligned body. These findings suggest that the perception of body location strongly influences feelings of ownership, emphasizing the role of spatial perception in body ownership. The manuscript is well-written, presenting the research clearly and concisely. The study is well-conducted, with rigorous methods that are appropriately designed and executed to address the research questions. The conclusions are sound and well-supported by the data. Overall, I found the work to be of high quality, and I have only minor comments and suggestions to improve the clarity or detail in some sections.

The inclusion of SCR to the virtual threat as a physiological measure is a valuable aspect of this study, providing an implicit indicator of emotional and autonomic responses during bodily illusions. This strengthens the findings and adds depth to the understanding of embodiment. However, I recommend that the authors expand the literature review surrounding the use of SCR in bodily illusions. This would further strengthen the rationale for its use and provide a more comprehensive context, highlighting the relevance of physiological measures in studying ownership and disownership phenomena.

Furthermore, the use of a virtual threat, (stabbing knife) directed at the virtual bodies, is an impactful element of the study. However, it would benefit from further clarification within the context of the bodily illusion manipulation. The authors should provide more detail on how this virtual threat is integrated into the experimental design and its specific role in measuring embodiment, disownership, and emotional responses. A clearer rationale for using the stabbing knife would help to better understand its contribution to the overall findings, particularly in relation to the skin conductance and subjective measures.

The hypothesis that FBI (compared to the RHI) may depend on the integration of proprioceptive information from the entire body, thereby making the initial visual exposure stage of the illusion less effective in establishing the overall illusion is very plausible tome and presents a compelling avenue for exploration. To strengthen this idea, the authors should provide more substantial evidence or theoretical backing that highlights the role of proprioceptive information in the FBI. Including relevant studies or empirical data that examine the interplay between proprioception and visual input in establishing bodily illusions would greatly enhance the discussion.

I would like to encourage the authors to elaborate further on how their findings can be applied within the context of current clinical neuroscience and rehabilitation practices. Specifically, it would be beneficial to discuss the implications of their results for therapeutic interventions aimed at individuals with body image disturbances. Expanding on this aspect can certainly provide valuable insights for practitioners in the field and highlight the broader significance of their work.

6. PLOS authors have the option to publish the peer review history of their article (what does this mean?). If published, this will include your full peer review and any attached files.

Reviewer #1: **Yes: **Bruno Herbelin

Reviewer #2: No

---

## [Author Response · Author response to Decision Letter 0]

14 Oct 2024

Dear Editor,

We greatly appreciate the opportunity to revise our manuscript titled “I am where I believe my body is: the interplay between body spatial prediction and body ownership.” We have carefully addressed the points raised by reviewers. Specifically, we clarified the terminology related to body-part location and improved the description of experimental conditions. Moreover, we expanded the discussion section according to the reviewers’ requests, incorporating additional references to support our findings and including clinical implications.

We hope you may find the current version of the manuscript improved. The modified parts in the manuscript are in red (see “Revised Manuscript with Track Changes” file).

Below are our point-by-point responses to the comments:

Reviewers' comments:

Reviewer's Responses to Questions

Comments to the Author

1. Is the manuscript technically sound, and do the data support the conclusions?

Reviewer #1: Yes

Reviewer #2: Yes

2. Has the statistical analysis been performed appropriately and rigorously? 

Reviewer #1: Yes

Reviewer #2: Yes

3. Have the authors made all data underlying the findings in their manuscript fully available?

Reviewer #1: Yes

Reviewer #2: Yes

4. Is the manuscript presented in an intelligible fashion and written in standard English?

Reviewer #1: Yes

Reviewer #2: Yes

We thank the reviewers for their positive feedback.

5. Review Comments to the Author

Reviewer #1: Overall, this manuscript is well written, methodologically strong, and scientifically interesting.

We thank Reviewer 1 for their positive comments.

The main issue I have with the manuscript is in the possible misinterpretation of the term ‘body location’, which is used here to speak of the location of body parts (hand or legs), while it is probably most often understood as the location of the body in space, i.e. self-location. This leads to possible misinterpretation of the results and of the experimental manipulation. Please ensure to rephrase the manuscript with term that avoids this ambiguity.

e.g. l 224: “toward the perceived body’s location”: this should read “towards the perceived hand location”. Please ensure this is the case throughout the manuscript (e.g. l 383, 390)

We thank the reviewer for the valuable observation. We acknowledge the concern that this term may be misinterpreted as referring to the overall position of the body in space, rather than the intended reference to the location of specific body parts (e.g., hand or legs).

To address this point, we have carefully revised the manuscript to eliminate any ambiguity.

Lines 247-249:

“Specifically, participants were required to perform a ballistic movement from the resting position of their right hand towards their left hand/leg and to point the virtual ray-cast toward the perceived location of the hand/leg as accurately as possible.”

Lines 254-257:

“A negative value of the Proprioceptive Drift means that the perceived hand/leg position is shifted to the right relative to their actual position. Conversely, a positive value indicates that the perceived hand/leg position is shifted to the left than the real body-part position.”

Lines 260-262:

“Given our prediction that embodiment depends on the body’s location prediction, we wanted to consider if participants predicted their body-part to be where the virtual body-part was and their confidence in that prediction”.

Lines 305-307:

“A negative value of the Proprioceptive Drift means that the body-part is perceived shifted to the right relative to the actual position. Conversely, a positive value indicates that the body-part is perceived as shifted to the left.”

Lines 411-412:

“As expected, after both illusions, a recalibration of the position of the hand/leg in space toward the shifted virtual body is observed.”

Lines 416-417:

“Moreover, a leftward bias emerged in estimating the hand/leg position, as if the body-part were perceived further from the body midline.”

Furthermore, in the discussion section, we have revised the term “body location” to “body-part location” when necessary to enhance clarity (marked in the text in red).

The description of the experimental conditions for experiment 1pp-FBI could benefit from a clarification.

L 135 says “left hand or legs were shifted towards the body's midline” : this describes a condition where there would be a lateral translation of the legs (like for the RHI).

However, the figure 1 shows that the “dislocation” condition is a difference in body posture (joint knees vs. open legs). This is a very important as the difference between conditions of the 1pp-FBI is therefore driven by the proprioception of legs postures (not a judgement of self-location in space).

We appreciate the reviewer’s comment. We recognize that the dislocation condition in the 1pp-FBI involves a difference in leg posture. However, we chose this manipulation as it was the only means to closely replicate the lateral shift typically employed in the RHI in the most ecologic manner. We also acknowledge that proprioceptive information related to leg posture may play a more significant role in this context. Thus, we have added this clarification to the text: 

Lines 141-143:

“Specifically, while a lateral shift to the right can be easily implemented for the hand manipulation, a lateral shift of the leg position requires a change in legs’ posture (i.e., transitioning from open to joint legs).”

Lines 584-587:

“Crucially, a lateral shift of the leg position in our manipulation requires a change in leg posture (i.e., open legs vs. joint legs), emphatising the difference between the two leg positions and increasing the weight of proprioceptive information.”

L494 “suggesting that simply viewing a fake body spatially superimposed to my body can induce ownership toward it” This is a very interesting observation that could be described more and related to the concept of ‘break in embodiment’ described in the literature.

We appreciate the reviewer’s suggestion. We have expanded the discussion to relate our findings to the concepts of the “Self-Avatar Follower Effect” and “breaks in embodiment” (Gonzalez-Franco et al., 2020; Porssut et al., 2023; Boban et al., 2024). 

Lines 522-528:

“Additionally, for the vRHI, ownership levels were high even in the misaligned condition prior to visuo-tactile stimulation. This aligns with the Self-Avatar Follower Effect [35-37], where an incongruence between real and virtual bodies can initially promote embodiment sensations, though embodiment may diminish if the mismatch persists or increases, potentially leading to a break in embodiment [36, 37]. In our study, ownership remained intact despite the mismatch, possibly due to the dominance of visual cues, with no break in embodiment. Furthermore, the multisensory congruence during the visuo-tactile stimulation further enhanced the sense of embodiment.”

The SCR data show a difference for a threat to the hand vs. leg (l509): it is however not discussed that this difference could be due to the perceived distance to the threat (threat to the hand being closer to the view than threat to the leg).

We thank the reviewer for the insightful interpretation of our result. We revised the manuscript by adding this point (lines 551-553):

“Another possible interpretation is that the perceived distance to the threat may influence the observed difference in arousal response. Specifically, a threat to the hand may be perceived as closer and, therefore, more alarming than a threat to the leg, potentially resulting in a stronger SCR.”

Also, related to the discussion on the perception of limb location (l533), an interpretation that could be discussed is the difference in distance perception in VR compared to reality (often reported as the difficulty to estimate distance, but in general to a distortion of distance perception in VR compared to reality).

We appreciate the Reviewer’s suggestion. We have revised the manuscript to incorporate this aspect (lines 589-593):

“Moreover, it is important to note that distance perception in VR can differ from real-world perception [47-49] with various factors contributing to these discrepancies [50]. Thus, it is possible that the immersive nature of VR and the way the body is represented within the virtual environment may cause participants to perceive the virtual limb’s position differently than they would in a real-world setting.”

Please ensure the English terminology is appropriate: e.g. ‘dislocation’ of an arm would rather be a medical injury with a dislocated joint.

We thank the reviewer for raising this point. We apologize if our terminology was inappropriate. We have replaced the term 'dislocation' with more appropriate terminology to better convey our meaning. The two location conditions will now be referred to as “Aligned” (co-location) when the virtual and real body positions match, and “Misaligned” when the virtual body is shifted toward the right. 

Reviewer #2: This study investigates how our ability to predict our body's spatial position influences feelings of ownership and disownership using two illusion techniques: the vRHI and the first-person perspective 1pp-FBI. Participants viewed a virtual body aligned or misaligned with their own, followed by synchronous tactile stimulation and a stabbing event. SCR was measured as an implicit embodiment indicator, along with a Body Localization Task and a questionnaire. Results showed that both illusions increased ownership, but this was weaker in the misaligned 1pp-FBI. Disownership occurred only in the misaligned 1pp-FBI, especially when legs were misaligned. Additionally, participants recalibrated their perceived body position toward the virtual misaligned body. These findings suggest that the perception of body location strongly influences feelings of ownership, emphasizing the role of spatial perception in body ownership. 

The manuscript is well-written, presenting the research clearly and concisely. The study is well-conducted, with rigorous methods that are appropriately designed and executed to address the research questions. The conclusions are sound and well-supported by the data. Overall, I found the work to be of high quality, and I have only minor comments and suggestions to improve the clarity or detail in some sections.

We thank Reviewer 2 for their positive comments.

The inclusion of SCR to the virtual threat as a physiological measure is a valuable aspect of this study, providing an implicit indicator of emotional and autonomic responses during bodily illusions. This strengthens the findings and adds depth to the understanding of embodiment. However, I recommend that the authors expand the literature review surrounding the use of SCR in bodily illusions. This would further strengthen the rationale for its use and provide a more comprehensive context, highlighting the relevance of physiological measures in studying ownership and disownership phenomena.

We thank the reviewer for the suggestion. We have revised the manuscript to include a broader discussion on the role of SCR (lines 88-94):

“Both explicit (i.e., Embodiment Scale; [11]) and implicit (i.e., Skin Conductance Responses; [14, 15]) measures were used to assess ownership and disownership. SCR captures autonomic responses to perceived threats against an embodied fake or virtual body. When the fake or virtual body is perceived as part of one’s own, threats to it elicit emotional and defensive reactions similar to threats to the real body [16]. Previous studies showed that threatening an embodied fake or virtual hand increases SCR significantly, indicating heightened emotional arousal [17, 18].”

Furthermore, the use of a virtual threat, (stabbing knife) directed at the virtual bodies, is an impactful element of the study. However, it would benefit from further clarification within the context of the bodily illusion manipulation. The authors should provide more detail on how this virtual threat is integrated into the experimental design and its specific role in measuring embodiment, disownership, and emotional responses. A clearer rationale for using the stabbing knife would help to better understand its contribution to the overall findings, particularly in relation to the skin conductance and subjective measures.

We thank the reviewer for the valuable feedback. In our study, the stabbing knife serves as a means of eliciting physiological and emotional responses directly linked to embodiment sensations elicited. This threat stimulus (i.e., the stabbing knife) was carefully chosen based on previous literature, proven to evoke a clear and measurable SCR, an indicator of emotional arousal and physiological reactivity (e.g., Ma & Hommel,2013; Petkova et al., 2011; Tieri et al., 2015).

In lines 207-215, we have better described how the virtual threat is integrated into the experimental design:

“Upon completion of the multisensory stimulation, a virtual knife appeared and stabbed the virtual left hand or leg. Specifically, after exactly 120 seconds, the experimenter pressed a button to trigger the appearance of the knife. The timing of the knife presentation was carefully coordinated to occur immediately after the visuo-tactile stimulation ended, ensuring that participants remained focused on the virtual body without shifting their attention elsewhere. The knife stabbed the virtual hand in the back, while the leg was stabbed in the thigh. The site of the stabbing corresponded to the prior location of the stick during the visuo-tactile stimulation. The total duration of the appearance and stabbing was approximately 1.5 seconds.” 

We have better described the specific role in measuring embodiment, disownership, and emotional responses:

- Lines 148-154: 

“After two minutes of visuo-tactile stimulation, a virtual knife appeared and stabbed the fake virtual body. The skin conductance level (SCL) was collected throughout the procedure to capture and monitor the sympathetic response to the virtual threat, according to the illusion and location condition. The stabbing knife served to elicit physiological and emotional responses directly linked to the participant’s sense of ownership or disownership. The threatening stimulus (i.e., the stabbing knife) was chosen based on previous literature, proven to evoke a clear and measurable SCL, an indicator of emotional arousal and physiological reactivity [22-24].”

- Lines 234-240: 

“The SCR P-P following the virtual threat is used as an indicator of the degree of embodiment: higher levels of SCR are typically associated with stronger feelings of ownership over the virtual body, as the virtual body is perceived as part of one’s body and threats to it would elicit emotional reactions 

---

## [Decision Letter · Decision Letter 1]

8 Nov 2024

I am where I believe my body is: the interplay between body spatial prediction and body ownership

PONE-D-24-08955R1

Dear Dr. Tosi,

We’re pleased to inform you that your manuscript has been judged scientifically suitable for publication and will be formally accepted for publication once it meets all outstanding technical requirements.

Kind regards,

Prof. Jane Elizabeth Aspell, PhD

Academic Editor

PLOS ONE

Additional Editor Comments (optional):

Reviewers' comments:

Reviewer's Responses to Questions

**Comments to the Author**

1. If the authors have adequately addressed your comments raised in a previous round of review and you feel that this manuscript is now acceptable for publication, you may indicate that here to bypass the “Comments to the Author” section, enter your conflict of interest statement in the “Confidential to Editor” section, and submit your "Accept" recommendation.

Reviewer #1: All comments have been addressed

Reviewer #2: All comments have been addressed

2. Is the manuscript technically sound, and do the data support the conclusions?

Reviewer #1: Yes

Reviewer #2: Yes

3. Has the statistical analysis been performed appropriately and rigorously? 

Reviewer #1: Yes

Reviewer #2: Yes

4. Have the authors made all data underlying the findings in their manuscript fully available?

Reviewer #1: Yes

Reviewer #2: Yes

5. Is the manuscript presented in an intelligible fashion and written in standard English?

Reviewer #1: Yes

Reviewer #2: Yes

6. Review Comments to the Author

Reviewer #1: Thank you for the improvements to the manuscript which make it clearer and address my comments. I have no further reserve for publishing this nice work.

Reviewer #2: The authors have addressed all my comments thoroughly and have done an excellent, detailed job in revising the manuscript. I am fully satisfied with the review.

7. PLOS authors have the option to publish the peer review history of their article (what does this mean?). If published, this will include your full peer review and any attached files.

Reviewer #1: **Yes: **Bruno Herbelin

Reviewer #2: No

---

## [Editor Report · Acceptance letter]

14 Nov 2024

PONE-D-24-08955R1 

PLOS ONE

Dear Dr. Tosi, 

I'm pleased to inform you that your manuscript has been deemed suitable for publication in PLOS ONE. Congratulations! Your manuscript is now being handed over to our production team.

Kind regards, 

on behalf of

Prof. Jane Elizabeth Aspell 

Academic Editor

PLOS ONE